# Based on Multi-Activity Integrated Strategy to Screening, Characterization and Quantification of Bioactive Compounds from Red Wine

**DOI:** 10.3390/molecules26216750

**Published:** 2021-11-08

**Authors:** Yuye Gao, Xie-an Yu, Bing Wang, Guo Yin, Jue Wang, Tiejie Wang, Kaishun Bi

**Affiliations:** 1School of Pharmacy, Shenyang Pharmaceutical University, Shenyang 110016, China; 13002480316@163.com; 2NMPA Key Laboratory for Quality Research and Evaluation of Traditional Chinese Medicine, Shenzhen Institute for Drug Control, Shenzhen 518057, China; yuxieanalj@126.com (X.-a.Y.); wangbing@szidc.org.cn (B.W.); ayinguoa@126.com (G.Y.); wangjuepha@126.com (J.W.)

**Keywords:** red wine, multi-activity integrated strategy, active components, coronary heart disease

## Abstract

According to French Paradox, red wine was famous for the potential effects on coronary heart disease (CHD), but the specific compounds against CHD were unclear. Therefore, screening and characterization of bioactive compounds from red wine was extremely necessary. In this paper, the multi-activity integrated strategy was developed and validated to screen, identify and quantify active compounds from red wine by using ultra high performance liquid chromatography-fraction collector (UHPLC-FC), ultra fast liquid chromatography-quadrupole-time-of-flight/mass spectrometry (UFLC-Q-TOF/MS) and bioactive analysis. UHPLC-FC was employed to separate and collect the components from red wine, which was further identified by UFLC-Q-TOF/MS to acquire their structural information. Furthermore, the active fractions were tested for antioxidant activity, inhibitory activity against thrombin and lipase activities in vitro by the activity screening kit. As the results, there were 37 fractions had antioxidant activity, 22 fractions had thrombin inhibitory activity and 28 fractions had lipase inhibitory activity. Finally, 77 active components from red wine were screened and 12 ingredients out of them were selected for quantification based on the integration of multi-activity. Collectively, the multi-activity integrated strategy was helpful for the rapid and effective discovery of bioactive components, which provided reference for exploring the health care function of food.

## 1. Introduction

As a nutritious drink recommended by World Health Organization (WHO), red wine was widely recognized for its role in health care. According to Ben Cao Gang Mu (Compendium of Materia Medica) record, the famous classic book of traditional Chinese medicine, red wine could treat cold coagulation and blood stasis. Modern pharmacological studies showed that red wine had the function of antioxidant, anticoagulant and lipid-lowering [1,2], which can prevent cardiovascular and cerebrovascular diseases such as coronary heart disease (CHD) [3,4].

It was widely believed that the mechanisms of CHD were the result of the interaction of many complex factors, among them the “response to injury hypothesis” had been recognized by researchers [5,6], which stated that the initial damage was the arterial endothelium, further leading to endothelial dysfunction [7]. Indeed, the occurrence of CHD was caused by many factors such as oxidative stress response, hyperlipidemia and platelet aggregation [8]. There were three main methods to treat CHD including pharmacotherapy, interventional therapy and operation [9,10]. Although these methods hold the potential effects for CHD, these methods would also lead some side effects such as common complications after coronary intervention included bleeding, pulmonary embolism, infection and so on. Besides, long term use of some medicines as nitroglycerin would also generate drug resistance and migraine pain [11]. Therefore, prevention CHD has become the trend of medical development.

The famous French paradox proved that red wine can prevent CHD [3]. Accumulated studies demonstrated that extracts from red wine possessed antioxidant activity, anti-coagulation and lipid-lowering efficacy, which corresponded to the pathogenesis of CHD. Traditional methods for isolating and evaluating components were most carried out on animal and cell models, which were time-consuming, labor-intensive, expensive and hardly to be used for the direct screening [12]. The developed methods for screening active compounds were extracted, the single component from sample and evaluated the biological activity by pharmacological means. Animal model was the most important screening manner in the part of bioactive components evaluation, which can clearly response to the efficacy of ingredients in the whole body. However, the method was high cost and low efficiency. Besides, compared with the animal model, cellular model was relatively cheap and convenient, but it had higher requirement on instrumentation [13]. Therefore, it was urgent to establish a fast, portable and accurate method to screen and identify the bioactive ingredients from red wine. Ultra high performance liquid chromatography (UHPLC) was used to separate the components of food with appropriate chromatographic conditions and the fraction collector was used to enrich compounds automatically in the fixed-time interval according to the retention time in the UHPLC chromatogram. Compared with HPLC, UHPLC exhibited higher separation and narrower chromatographic peak. At the same time, ACQUITY UPLC column (HSS T3, 1.8 μm, 2.1 mm × 100 mm, Waters, Ireland) could withstand 100% water mobile phase and the particle size was only 1.8 microns, which was adapted to separation of complex samples especially for the strong polar compounds from red wine. Indeed, some researches showed that ultra high performance liquid chromatography-fraction collector (UHPLC-FC) could quickly separate and enrich compounds. Moreover, ultra fast liquid chromatography-quadrupole-time-of-flight/mass spectrometry (UFLC-Q-TOF/MS) was used to identify the components for obtaining structural information [12]. Accordingly, integrated the UHPLC-FC and UFLC-Q-TOF/MS could be used for quickly separating, enriching and identifying the potential active compounds from complex samples.

In this study, UHPLC-FC was used to separate, collect and enrich the fractions from red wine. Then antioxidant activity, inhibitory activity against thrombin and lipase activities of all fractions were performed for evaluating activities. Furthermore, UFLC-Q-TOF/MS was used to identify the components in fractions from red wine (Figure 1). As the results, 77 potential active components from red wine were screened and 12 ingredients out of them were selected for quantification based on integration of the multi-activity. Accordingly, the multi-activity integrated strategy was significant for the rapid and effective discovery of the potential bioactive components from red wine to prevent CHD, which provided reference for exploring the health care function of food.

## 2. Result and Discussion

### 2.1. Establishment of Multi-Activity Integrated Strategy

In the study, an ultra high performance liquid chromatography-photo-diode array-fraction collector (UHPLC-PDA-FC) system was used to establish multi-activity integrated strategy for screening and evaluating the potentially bioactive components from red wine. The sample was divided into sixty fractions (60 s per fraction) by UHPLC-PDA-FC. Sixty fractions were used to screen the antioxidant activity, thrombin inhibitory activity and lipase inhibitory activity. The multi-activity integrated chromatogram consisted of chromatogram and multi-targets integrated column diagram of antioxidant activity, inhibitory activity of thrombin and lipase were established (Figure 2) and the detail data was listed in Table 1. The multi-activity integrated strategy could discover bioactive components rapidly and efficiently, which could be used for the identification and screening the active compounds from nutraceutical. According to the activities results, the activities were higher at 20 to 30 min, indicating that these kinds of ingredients may be the main active compounds from red wine.

### 2.2. Identification of 94 Components from Red Wine

The components from red wine with antioxidant activity or inhibitory potential against thrombin and lipase activities were analyzed by UFLC-Q-TOF/MS to obtain structural information. The molecular compositions of ingredients were identified according to the literature and exact molecular weight. The peak number, retention time, MS/MS fragmentation ions information and identification of 94 components from red wine were listed in Table 2. The extracted ion chromatogram of positive and negative ions was illustrated in Figure 3. Peak 1 gave the mass spectrum ion at 116.0702 (C_5_H_9_NO_2_) with MS/MS fragment of 70.0647 (C_4_H_7_N) [M+H-HCOOH]^+^, which was detected to be D-proline [14,15,16]. Peaks 11 (C_4_H_9_N) [M+H-HCOOH]^+^ and 14 (C_5_H_3_N) [M+H-HCOOH]^+^ were based on the same regular. Therefore, peaks 11 and 14 were identified as L-valine (C_5_H_11_NO_2_) [14] and niacin (C_6_H_5_NO_2_) [17]. Peak 2 had the mass spectrum ion at 175.1189 (C_6_H_14_N_4_O_2_) with MS/MS fragment of 70.0652 [M+H-HCOOH-2NH_2_-CN]^+^, which was detected to be L-arginine [14]. Peak 3 had a molecular weight of 179.0562 (C_6_H_12_O_6_) with fragments of 99.0087 (C_4_H_4_O_3_) [M-H-3H_2_O-C_2_HO]^−^, 71.0137 (C_3_H_4_O_2_) [M-H-3H_2_O-C_2_HO-CO]^−^ and 59.0139 (C_2_H_4_O_2_) [M-H-3H_2_O-C_2_HO-CO-CH_2_]^−^. Therefore, it was identified as D-glucose. Peak 4 had mass spectrum ion at 181.072 (C_6_H_14_O_6_) with MS/MS experiment gave fragmentation information for 83.0143 [M-H-C_3_H_2_O_3_]^−^ and 71.0140 [M-H-C_3_H_6_O_3_-H_2_O]^−^, which could be identified as D-mannitol [18]. Peak 5 had major ions at m/z 191.0199 (C_6_H_8_O_7_) and the MS/MS fragments were 85.0296 (C_4_H_6_O_2_) [M-H-2CO_2_-H_2_O]^−^ and 59.0134 (C_3_H_8_O) [M-H-3CO_2_]^−^. Therefore, it was identified as citric acid [19]. Peak 6 was identified as tartaric acid by its characteristic fragmentation pattern, which MS/MS spectra ions m/z 105.0217 (C_3_H_6_O_4_) [M-H-CO_2_]^−^ and 87.0085 (C_3_H_4_O_3_) [M-H-CO_2_-H_2_O]^−^ were observed [20]. By the similar regular pattern, peak 40 (C_9_H_8_O_3_) had the MS/MS fragment of m/z 119.0502 (C_8_H_8_O) [M-H-CO_2_]^−^. Therefore, Peak 40 was identified as 2-hydroxycinnamic acid [21]. Peak 7, the molecular weight was 138.0548 (C_7_H_7_NO_2_) and the fragment ions observed included m/z 120.0433 (C_6_H_7_N) [M+H-H_2_O]^+^, 78.0339 (C_6_H_6_) [M+H-HCOOH-NH_2_]^+^, which was identified to be 4-aminobenzoic acid [22]. Peak 8 gave a mass spectrum ion at 341.1088 (C_12_H_22_O_11_) with MS/MS fragments of 161.0459 (C_6_H_9_O_5_) [M-H-C_6_H_11_O_5_-H_2_O]^−^ and 71.0137 (C_3_H_4_O_2_) [M-H-C_9_H_14_O_8_-H_2_O]^−^, which was detected to be trehalose [23]. The molecular weight of peak 9 was 149.0455 (C_5_H_10_O_5_). In the MS/MS experiment, the predominant ions appeared at 75.0221 (C_3_H_7_O_2_) [M-H-C_2_H_3_O_3_]^−^ and 59.0133 (C_2_H_2_O_2_) [M-H-C_3_H_8_O_2_]^−^. By comparing the characteristic fragmentation pattern with the reported in references, it can be identified as D-ribose. Peaks 10 (C_4_H_6_O_5_) was identified as L-malic acid [24] which the H_2_O was removed at first, and then the CO_2_ was removed. MS/MS fragment of L-malic acid was 115.0018 (C_4_H_4_O_4_) [M-H-H_2_O]^−^ and 71.0137 (C_3_H_4_O_2_) [M-H-H_2_O-CO_2_]^−^. Peaks 12 (C_7_H_10_O_5_) was identified as shikimic acid. MS/MS fragment was 137.0225 (C_7_H_6_O_3_) [M-H-2H_2_O]^−^. The molecular weight of peak 13 was 150.0586 (C_5_H_11_NO_2_S) and ion m/z 74.0243 (C_3_H_8_O_2_) [M+H-NH_3_-C_2_S]^+^ was observed in MS/MS spectra, which could be identified as L-methionine [16]. Peak 15 had a molecular weight of 117.0192 (C_4_H_6_O_4_), which produced the MS/MS ions as 99.9254 (C_3_H_6_O_2_) [M-H-CO_2_]^−^ and 73.0291 (C_3_H_4_O) [M-H-CO_2_-H_2_O]^−^. Hence, it can be identified as succinic acid [25]. Peak 33 had the same fragment (73.0273 [M-H-C_3_H_5_O_2_]^−^) with peak 15. The other fragment was 55.0191 [M-H-C_3_H_5_O_2_-H_2_O]^−^, which can be identified as dimethyl succinate [25]. Peaks 18, 21, 23, 24, 25, 29, 31, 32, 33, 39, 42, 43, 49, 50, 52, 55, 56, 60 and 61 were in the similar regular pattern with peak 15, the parent ion was removed one or more CO_2_ and H_2_O. Peaks 18 (125.0244 [M-H-CO_2_]^−^, 107.0156 [M-H-CO_2_-H_2_O]^−^ and 79.0189 [M-H-CO_2_-H_2_O-CO]^−^), 21 (109.0298 [M-H-CO_2_]^−^ and 91.0298 [M-H-CO_2_-H_2_O]^−^), 23 (135.0560 [M-H-CO_2_]^−^ and 89.0194 [M-H-CO_2_-CH_3_CO]^−^), 24 (135.0454 [M-H-CO_2_]^−^ and 117.0351 [M-H-CO_2_-H_2_O]^−^), 25 (109.0298 [M-H-CO_2_]^−^ and 91.0298 [M-H-CO_2_-H_2_O]^−^), 29 (93.0340 [M-H-CO_2_]^−^), 31 (163.0400 [M-H-2CO_2_]^−^ and 119.0503 [M-H-3CO_2_]^−^), 32 (119.0498 [M-H-CO_2_]^−^ and 93.0347 [M-H-CO_2_-H_2_O]^−^), 33 (163.0400 [M-H-2CO_2_]^−^ and 119.0498 [M-H-3CO_2_]^−^), 39 (108.0212 [M-H-CO_2_-CH_3_]^−^), 42 (193.0490 [M-H-2CO_2_-C_2_H_5_O]^−^ and 117.0341 [M-H-3CO_2_-C_2_H_5_O-CH_3_-H_2_O]^−^), 43 (135.0449 [M-H-CO_2_]^−^), 49 (119.0527 [M-H-CO_2_]^−^ and 103.0565 [M-H-CO_2_-H_2_O]^−^), 50 (93.0470 [M-H-CO_2_]^−^ and 65.0473 [M-H-CO_2_-CO]^−^), 52 (123.0436 [M-H-CO_2_]^−^ and 108.0212 [M-H-CO_2_-H_2_O]^−^), 55 (119.0503 [M-H-CO_2_]^−^ and 93.0347 [M-H-CO_2_-H_2_O]^−^), 56 (123.0450 [M-H-CO_2_]^−^), 60 (257.0110 [M-H-CO_2_]^−^ and 229.0126 [M-H-CO_2_-CO]^−^) and 61 (109.0286 [M-H-CO_2_-C_2_H_2_]^−^) were the same regular, which were identified as gallic acid [20], protocatechuic acid [23], acetylsalicylic acid [26], caftaric acid [20], gentisic acid [23], 4-hydroxybenzoic acid [27], p-coutaric acid [28], m-coumaric acid [28], coutaric acid [28], vanillic acid [27], fertaric acid [28], caffeic acid [28], desaminotyrosine [29], 2-hydroxybenzoic acid [28], 3-hydroxy-4-methoxybenzoic acid [30,31], p-coumaric acid [28], homogentisic acid [23], ellagic acid [32] and dihydrocaffeic acid [33]. Peak 79 had the major first-order mass spectrum at m/z 270.0662 (C_11_H_12_O_4_). The MS/MS fragments were the same as peak 43 of 179.0433 [M-H-CH_3_]^−^ and 135.0446 [M-H-CH_3_-CO_2_]^−^. Therefore, it was identified as ethyl caffeate [28]. Peak 16 showed the mass spectrum ion at m/z 165.0547 (C_9_H_8_O_3_) and the product ion was 119.0479 [M+H-HCOOH]^+^, which could be identified as trans-4-hydroxycinnamic acid [20]. Peak 17 gave a mass spectrum ion at 138.0548 (C_8_H_11_NO) with the MS/MS fragments at 121.0616 [M+H-NH_3_]^+^ and 95.0561 [M+H-NH_3_-C_2_H_4_]^+^, which was detected to be tyramine [34,35]. Peak 19 showed the mass spectrum ion at 141.0182 with the MS/MS fragments at 95.0127 [M+H-HCOOH]^+^, which could be identified as coumalic acid. Peak 20 gave [M−H]^−^ ion at m/z 153.0558. In MS/MS mode, the produced ions were m/z 123.0085 [C_8_H_10_O_3_-CH_2_O]^−^ and 108.0221 [C_8_H_10_O_3_-CH_2_OH-CH_3_]^−^, which can be identified as hydroxytyrosol [36]. Peaks 22 and 37 were a type of isomer which yielded the same ion at 305.0246 [M−H]^−^ in the first-order mass spectrum. They had the same prominent ion at 125.0246 [C_15_H_14_O_7_-C_9_H_8_O_4_]^−^. They were speculated to be (−)-epigallocatechin [25] and (+)-gallocatechin [25] based on their retention times. Peak 26 had the [M−H]^−^ ion at m/z 137.0255. The product ion was 108.0213 [M-H-CHO]^−^. Therefore, peak 26 was identified as protocatechualdehyde. Peak 27 had the mass spectrum ion at 181.0506 (C_9_H_10_O_4_) with MS/MS fragment of 135.0453 [M-H-CH_3_-H_2_O]^−^, which was identified as homovanillic acid [34,37]. Peak 28 showed mass spectrum ion at 183.0299 (C_8_H_8_O_5_). The product fragment ions were 124.0234 [M-H-CO_2_-CH_3_]^−^ and 95.0136 [M-H-CO_2_-CH_3_-CHO]^−^. Hence, it was identified as methyl gallate [38,39]. Peak 30 gave [M−H]^−^ ion at m/z 353.0877 and MS/MS produced ions were at m/z 191.0616 [M-H-C_6_H_5_O_5_]^−^ and 135.0439 [M-H-C_6_H_5_O_5_-CO]^−^, it can be identified as chlorogenic acid [27,40]. Peak 35 was identified as benzoic acid by the MS spectra ion 121.0296 and MS/MS spectra ions m/z 92.0265 [M-H-CHO]^−^ and 108.0212 [M-H-CH]^−^. Peaks 36, 38, 46 and 57 had same molecular weight of C_30_H_26_O_12_, which illustrated that they were isomers. Base on the MS/MS spectra, peak 36 (407.0775 [M-H-C_8_H_8_O_3_-H_2_O]^−^, 289.0714 [M-H-C_15_H_13_O_6_]^−^), 38 (407.0768 [M-H-C_8_H_8_O_3_-H_2_O]^−^, 289.0709 [M-H-C_15_H_13_O_6_]^−^), 46 (425.0893 [M-H-C_8_H_8_O_3_]^−^, 407.0782 [M-H-C_8_H_8_O_3_-H_2_O]^−^, 289.0721 [M-H-C_15_H_13_O_6_]^−^) and 57 (425.0886 [M-H-C_8_H_8_O_3_]^−^, 407.0789 [M-H-C_8_H_8_O_3_-H_2_O]^−^, 289.0734 [M-H-C_15_H_13_O_6_]^−^) were detected, which could be identified as procyanidin B1 [41], procyanidin B3 [41], procyanidin B2 [41] and procyanidin B7 [41] by comparing the retention time of the reference standards and literatures. Peak 41 had a molecular weight of 289.0713 (C_15_H_14_O_6_) with fragments of 245.0843 [M-H-CO_2_]^−^, 203.0713 [M-H-H_2_O-C_3_O_2_]^−^ and 151.0401 [M-H-C_8_H_10_O_2_]^−^, therefore it was identified as catechin [23]. Based on the same fragment, peaks 44 (289.0735 [M-H-C_7_H_2_O_4_]^−^, 151.0386 [M-H-C_7_H_4_O_4_-C_8_H_10_O_2_]^−^), 51 (203.0710 [M-H-H_2_O-C_3_O_2_]^−^, 187.0396 [M-H-C_4_H_6_O_3_]^−^ and 151.0401 [M-H-C_8_H_10_O_2_]^−^), 58 (316.0225 [M-H-C_6_H_11_O_5_]^−^, 271.0249 [M-H-C_6_H_11_O_5_-CO_-_H_2_O]^−^ and 151.0034 [M-H-C_14_H_16_O_9_]^−^), 64 (285.0413 [M-H-C_6_H_11_O_5_]^−^ and 151.0041 [M-H-C_14_H_8_O_7_]^−^), 66 (301.0352 [M-H-C_6_H_10_O_5_]^−^, 255.0296 [M-H- C_6_H_10_O_5_-C_2_H_2_O]^−^ and 151.0034 [M-H-C_14_H_11_O_9_]^−^), 67 (330.0381 [M-H-C_6_H_11_O_5_]^−^, 315.0155 [M-H-C_6_H_11_O_5_-CH_3_]^−^, 271.0249 [M-H-C_6_H_11_O_5_-CH_3_-CO-H_2_O]^−^ and 151.0034 [M-H-C_15_H_18_O_9_]^−^), 68 (151.0020 [M-H-C_14_H_16_O_7_]^−^), 69 (285.0413 [M-H-C_6_H_11_O_5_]^−^ and 151.0041 [M-H-C_14_H_8_O_7_]^−^), 70 (301.0379 [M-H-C_6_H_10_O_5_]^−^ and 151.0039 [M-H-C_6_H_10_O_5_-C_7_H_6_O_2_-CO]^−^), 72 (301.0372 [M-H-C_6_H_10_O_5_]^−^, 255.0293 [M-H-C_6_H_10_O_5_-C_2_H_2_O]^−^ and 151.0016 [M-H-C_14_H_11_O_9_]^−^), 73 (245.0504 [M-H-H_2_O-2CO]^−^ and 151.0035 [M-H-C_8_H_6_O_4_]^−^), 75 (315.0530 [M-H-C_6_H_10_O_5_]^−^, 271.0255 [M-H-C_8_H_14_O_6_]^−^ and 151.0034 [M-H-C_15_H_18_O_8_]^−^), 83 (151.0034 [M-H-C_8_H_6_O_3_]^−^), 85 (316.0231 [M-H-CH_3_]^−^ and 151.0016 [M-H-C_9_H_8_O_4_]^−^), 86 (151.0034 [M-H-C_8_H_8_O]^−^), 88 (151.0034 [M-H-C_8_H_6_O_2_]^−^) and 91 (300.0270 [M-H-CH_3_]^−^ and 151.0016 [M-H-C_9_H_8_O_3_]^−^) were identified as the type of ion of 151.0037 (C_7_H_4_O_4_), which were the characteristic fragment of flavonoid components. Indeed, they were epicatechin gallate [23], epicatechin [23], myricetin-3-O-galactoside [23], quercitrin [28], quercetin-3-O-glucuronide [28], laricitrin-3-O-glucoside [37], luteolin-3’-glucoside [23], astilbin [38], isoquercitrin [22], astragalin [38], myricetin [21], isorhamnetin-3-O-glucoside [28], quercetin [22], laricitrin [37], naringenin [23], kaempferol [22] and isorhamnetin [28]. Peak 45 had the mass spectrum ion at 197.0454 (C_9_H_10_O_5_) with MS/MS ions at 182.0256 [M-H-CH_3_]^−^, 167.0014 [M-H-CH_3_-CO_2_]^−^, 138.0317 [M-H-CH_3_-CO_2_-CH_3_]^−^ and 123.0090 [M-H-CH_3_-2CO_2_-CH_3_]^−^, which can be identified as syringic acid [27]. Peak 47 produced [M−H]^−^ at m/z 325.0926 with the molecular formula C_15_H_18_O_8_. A loss of C_6_H_10_O_5_ in MS/MS was confirmed to be a hexose neutral loss. The other ions were 145.0294 [M-H-C_6_H_10_O_5_-H_2_O]^−^ and 117.0343 [M-H-C_6_H_10_O_5_-CO]^−^. Peak 47 was tentatively identified as p-coumaric acid glucoside [28]. Peak 48 had the major ions at m/z 189.0769 (C_8_H_14_O_5_) and the MS/MS fragment was 71.0503 [M-H-C_2_H_5_-CO_2_-C_2_H_5_O]^−^. Therefore, it was identified as diethyl malate [24]. Peak 53 was identified as ethyl gallate by its characteristic fragmentation pattern, which MS/MS spectra ions m/z 169.0142 [C_9_H_10_O_5_-C_2_H_4_]^−^ and 125.0239 [C_9_H_10_O_5_-C_2_H_4_-CO_2_]^−^ were observed [28]. Peak 54 had the MS ion of 493.1341 and the MS/MS spectra ions of m/z 331.0822 [M-H-C_6_H_10_O_5_]^−^ and 316.0553 [M-H-C_6_H_10_O_5_-CH_3_]^−^. Hence, peak 54 was tentatively identified as malvidin-3-O-glucoside [41]. By the similar regular pattern, peaks 62 (303.0510 [M-H-C_6_H_10_O_5_]^−^ and 316.0553 [M-H-C_6_H_10_O_5_-2H_2_O-CO]^−^), 65 (303.0510 [M-H-2C_6_H_10_O_5_]^−^), 74 (317.0648 [M-H-C_6_H_10_O_5_]^−^ and 302.0429 [M-H-C_6_H_10_O_5_-CH_3_]^−^), 78 (303.0767 [M-H-C_6_H_10_O_5_-CO_2_]^−^) and 80 (477.1196 [M-H-C_9_H_10_O_3_]^−^ and 331.0874 [M-H-C_9_H_10_O_3_-C_6_H_10_O_5_]^−^) were displayed in MS/MS spectra, which could be identified as delphinidin-3-O-glucoside [42,43], delphinidin-3,5-O-diglucoside [44], petunidin-3-O-glucoside [41], delphindin-3-O-(6-O-acetylglucoside) [44] and malvidin-3-O-(6-O-coumaroylglucoside) [44]. Peak 59 gave the [M−H]^−^ ion at the m/z 389.1242 with MS/MS spectra ions m/z 227.0994 [M-H-C_6_H_10_O_5_]^−^ and 143.0505 [M-H-C_6_H_10_O_5_-C_2_H_2_O-C_2_H_2_O]^−^. Peak 59 was tentatively identified as trans-piceid [23]. By the similar regular pattern, peaks 72 (227.0994 [M-H-C_6_H_10_O_5_]^−^ and 185.0606 [M-H-C_6_H_10_O_5_-C_2_H_2_O]^−^), 76 (185.0587 [M-H-C_6_H_10_O_5_-C_2_H_2_O]^−^ and 143.0501 [M-H-C_6_H_10_O_5_-C_2_H_2_O-C_2_H_2_O]^−^) and 84 (143.0504 [M-H-C_6_H_10_O_5_-C_2_H_2_O-C_2_H_2_O]^−^) were displayed in MS/MS spectra, which can be identified as cis-piceid [23], trans-resveratrol [23] and resveratrol [23] with their different retention times. Peak 63 showed the mass spectrum ion at 463.0884 (C_21_H_20_O_12_). The product fragment ions were 301.0379 [M-H-C_6_H_10_O_5_]^−^, 300.0277 [M-H-C_6_H_11_O_5_]^−^, 271.0245 [M-H-C_6_H_10_O_5_-CHO]^−^, 243.0291[M-H-C_6_H_10_O_5_-CHO_2_]^−^ and 151.0031 [M-H-C_6_H_10_O_5_-C_7_H_6_O_2_-CO]^−^. Therefore, it was identified as hyperoside [23]. Peak 77 gave a mass spectrum ion at 435.1300 (C_21_H_24_O_10_) with MS/MS fragments of 273.0769 [M-H-C_6_H_12_O_6_]^−^, 179.0349 [M-H-C_15_H_14_O_5_]^−^ and 167.0352 [M-H-C_6_H_12_O_6_-C_7_H_6_O]^−^, which was detected to be phloridzin [45]. Peak 81 had a molecular weight of 303.0497 (C_15_H_10_O_7_) with fragments of 285.0398 [M+H-H_2_O]^+^, 257.0456 [M+H-H_2_O-CO]^+^, 229.0492 [M+H-H_2_O-2CO]^+^ and 153.0188 [M+H-C_8_H_5_O_3_]^+^, therefore it was identified as morin [46]. Peak 82 had the mass spectrum ion at 303.0497 and MS/MS experiment gave fragmentations information at 229.0492 [M-H-CO-2H_2_O]^−^ and 153.0188 [M-H-C_8_H_6_O_3_]^−^,which can be identified as delphinidin [28]. By the similar regular pattern, peaks 87 (213.0559 [M-H-2H_2_O-CO] and 137.0233 [M-H-C_8_H_6_O_3_]) and 90 (229.0498 [M-H-2H_2_O-CO-CH_3_] and 153.0188 [M-H-CH_3_-C_8_H_6_O_3_]) were showed in MS/MS spectra, which can be identified as cyanidin [28] and petunidin [28]. Peak 89 was identified as syringetin with MS/MS spectra ions m/z 330.0385 [M-H-CH_3_]^−^ and 315.0153 [M-H-C_2_H_6_]^−^ [47]. Peak 92, the molecular weight was 193.1588 (C_13_H_20_O) and the fragment ion was observed at m/z 56.9419 [M+H-C_10_H_15_]^+^, which was identified as ionone [48]. Peak 93 had a molecular weight of 173.1536 (C_10_H_20_O_2_) with fragment of 76.0221 [M+H-HCOOH]^+^, therefore it was identified as ethyl caprylate (C_10_H_20_O_2_). The molecular weight of peak 94 was 195.1748. MS/MS ions of m/z 177.1634 [M+H-H_2_O]^+^ and 163.0394 [M+H-H_2_O-CH_2_]^+^ were observed, which revealed that it was theaspirane. Moreover, the MS spectrums of all compounds were shown in Appendix A (Appendix A).

### 2.3. Active Ingredients of Red Wine

The analytical data of active fractions with antioxidant activity, inhibitory activity of thrombin or lipase and 94 compounds of mass spectrometry were correlated according to retention times and peaks order. In all, 77 compounds were screened for the prevention and treatment of CHD. The corresponding relationship between fraction number and peak number of the potential active components was shown in Table 3. These active components were concentrated in phenolic acids and flavonoids. Phenolic acids were retained during 20–30 min, which had higher antioxidant and inhibitory activity against thrombin and lipase. Flavonoids were retained during 42–60 min, which had stronger antioxidant and inhibitory activity against lipase. Besides, five ensemble approaches were considered to screen the quantitative composition. The ingredients had multi-activity at the same time; the compounds had a strong single activity; the reference substance was easy to obtain; the activity of this ingredient has been reported many times in literature; the compound was abundant, respectively. The five ensemble approaches were applicated for selecting active ingredients to the quality evaluation of red wine. 63 compounds had multiple activities; 13 compounds had a strong single activity, which were protocatechuic acid, (+)-catechin gallate, (+)-epigallocatechin, acetylsalicylic acid, caftaric acid, chlorogenic acid, benzoic acid, procyanidin B7, trans-piceid, astragalin, cis-piceid, ethyl caffeate and malvidin-3-O-(6-O-coumaroylglucoside). According to the reference substance was easy to obtain and the activity of this ingredient has been reported many times in literature, a total of 19 compounds were screened, which were succinic acid, gallic acid, coumalic acid, procyanidin B1, (−)-epigallocatechin, vanillic acid, catechin, caffeic acid, syringic acid, epicatechin, p-coumaric acid, quercetin-3-O-glucuronide, isoquercitrin, isorhamnetin-3-O-glucoside, trans-resveratrol, quercetin, protocatechuic acid, (+)-catechin gallate and (+)-epigallocatechin. Finally, based on the content of the compound from red wine, 12 compounds were screened for further determination, which were gallic acid, coumalic acid, proanthocyanidin B1, catechin, caffeic acid, isoquercitrin, protocatechin, syringic acid, epicatechin, p-coumaric acid, isorhamnetin-3-O-glucoside and quercetin.

### 2.4. UHPLC Analysis

The UHPLC conditions were considered to obtain better chromatographic separation. The different concentrations of acid (0.03%, 0.05% and 0.1% trifluoroacetic acid, 0.05% formic acid, 0.05% acetic acid and pure water), column temperatures (25, 30 and 35 °C) and detection wavelength (210, 220, 254 and 280 nm) were optimized. By comparing resolutions and the peak shapes, the better separation was achieved when 0.1% trifluoroacetic acid was selected as mobile phase, the column temperature and flow rate were optimal at 30 °C and 0.3 mL min^−1^, respectively. The UHPLC chromatogram was shown in Figure 4.

### 2.5. Method Validation

The precision, linearity, stability, accuracy, LODs and LOQs were validated. The RSD values of intra-day and inter-day precision were all less than 2.2% and 4.8%, respectively. Overall, 12 components were stable within 24 h at room temperature, and RSDs were less than 2.9%. The calibration curves of the 12 compounds were of high correlation coefficient (r > 0.999) under the concentration ranges. The LODs were ranged from 0.04 to 0.75 μg/mL and LOQs were ranged from 0.12 to 2.0 μg/mL for the 12 compounds, respectively. The results were shown in (Table 4). As can be seen from Table 5, the recoveries of the 12 compounds were in the range of 83.4–106% and RSDs values were not more than 4.3%. All these values were found in an acceptable range, indicating that the method was accurate, reproducible and reliable for quantification of bioactive compounds from red wine.

### 2.6. Contents of the Bioactive Components from Red Wine

UHPLC was applied to analyze the bioactive components in triplicates from red wine. The results showed that among the bioactive compounds, gallic acid was in the highest amounts of 46.94 μg/mL. The contents of coumalic acid (18.93 μg/mL), proanthocyanidin B1 (14.19 μg/mL), catechin (19.37 μg/mL), caffeic acid (12.05 μg/mL) and isoquercitrin (10.21 μg/mL) were in the range of 10.21–19.37 μg/mL. Protocatechin (4.768 μg/mL), syringic acid (9.87 μg/mL), epicatechin (7.148 μg/mL), p-coumaric acid (8.18 μg/mL), isorhamnetin-3-O-glucoside (6.460 μg/mL) and quercetin (5.353 μg/mL) were in low contents. Contents of 35 batches red wine were shown in (Figure 5). These components may be considered to be the main bioactive compounds from red wine and had the antioxidant activity, inhibitory activities of and lipase.

### 2.7. Confirmation the Activity of the 12 Compounds

In order to verify that the compounds in the obtained fractions, 12 active compounds mixture were contrasted for red wine in terms of antioxidant activity, thrombin inhibitory activity and lipase inhibitory activity. The results indicated that inhibition of selected 12 compounds had reached almost 90% of red wine. The inhibition ratios of 12 compounds mixture and red wine were displayed in Figure 6. Moreover, the difference between inhibition of red wine and 12 compounds mixture were also shown in Figure 6. Comparing the inhibition ratio of red wine and 12 compounds which concentrations were the same as red wine, it was shown that the inhibition ratios of 12 compounds were similar to red wine, which was verified that the screening 12 compounds could reflect the total activity of red wine in terms of antioxidant activity, thrombin inhibitory activity and lipase inhibitory activity. Accordingly, the blindness of selecting quantitative component was avoided, which could further provide reference for screening and quantification the active ingredients from nutraceutical and traditional Chinese medicine.

### 2.8. The Application of Multi-Activity Integrated Strategy

Red wine contained many ingredients, which had different activities such as antioxidant, anticoagulant and lipid-lowering. CHD was the results of the interaction of many complex factors. Hence, the multi-activity of red wine was corresponding to the multiple pathogenesis of CHD. Indeed, the multi-activity integrated strategy of red wine was successfully established to quality evaluation of red wine based on the screened 12 bioactive compounds to prevent CHD. Additionally, the multi-activity integrated strategy may help to discover bioactive components rapidly and efficiently, which provided reference for exploring active ingredients in food.

## 3. Materials and Methods

### 3.1. Reagents and Chemical

HPLC-grade Acetonitrile (ACN) and methanol (MeOH) were purchased from Merck (Darmstadt, Germany). Trifluoroacetic acid (TFA) was obtained from Guangzhou Chemical Reagent Factory (Guangzhou, China). Reference Standards including gallic acid (98.1%), catechin (99.2%), caffeic acid (99.7%), epicatechin (99.7%), isoquercitrin (98.0%) and quercetin (98.0%) were provided from National Institutes for Food and Drug Control (Beijing, China). Coumalic acid (98.0%), proanthocyanidin B1 (98.0%) and syringic acid (98.0%) were purchased from Chengdu Chroma-Biotechnology Co., Ltd. (Sichuan, China). P-coumaric acid (98.0%) and isorhamnetin-3-O-glucoside (98.0%) were offered by Shanghai Tauto Biotech Co., Ltd. (Shanghai, China). The water used in this study was purified by a Milli-Q water purification system (MA, USA). Total Antioxidant Capacity Assay Kit (ABTS) was supplied from Beyotime (Shanghai, China). Thrombin Inhibitor Screening Kit (Fluorometric) was obtained from Biovision (San Francisco, CA, USA). Lipase (TypeII, L3126) was purchased from Solarbio Science Co. Ltd. (Beijing, China) and 4-Methylumbelliferyl Oleate (4-MUO) was provided by Sigma-Aldrich (St. Louis, MO, USA). Red wine (2018 vintage, the central valley of Chile, alcoholic degree 13.0% (*v/v*)) was made of cabernet sauvignon, which was obtained from Ole’ Boutique Supermarket (Shenzhen, Guangdong, China).

### 3.2. Sample and Standard Solution Preparation

The red wine was analyzed immediately after the bottle opening. The sample (2 mL) was filtrated through a 0.22 μm polyvinylidene fluoride membrane. Appropriate amounts of gallic acid, catechin, caffeic acid, epicatechin, isoquercitrin, coumalic acid, proanthocyanidin B1, syringic acid, p-coumaric acid and isorhamnetin-3-O-glucoside were precisely weighed and dissolved with methanol to prepare standards solutions of 1 mg/mL. All the samples and standard solutions were stored at 4 °C in the dark for analysis.

### 3.3. Preparation of the Red Wine Fractions

Fraction collector (BSZ-100, Shanghai Jiapeng Technology Instrument, Shanghai, China) was used to prepare fractions. The red wine sample was injected into the UHPLC system for separation. Then fractions were collected every 60 s by setting the fraction collector and evaporated to dryness by nitrogen gas. The residues were dissolved by methyl alcohol:water (1:1).

### 3.4. UHPLC Analysis

The UHPLC analysis was performed using an Dionex ultra high performance liquid chromatograph (Dionex UltiMate 3000, Thermo, MA, USA) equipped with a Dionex UltiMate 3000 pump, a Dionex UltiMate 3000 autosampler and a Dionex UltiMate 3000 diode array detector. The chromatographic separation was achieved on an ACQUITY UPLC column (HSS T3, 1.8 μm, 2.1 mm× 100 mm, Waters, Ireland) [49] at 30 °C. The mobile phase was composed of water containing 0.1% TFA (A) [50] and methanol (B) at a flow rate of 0.3 mL/min. The elution program was conducted as follows: 0–15 min at 0–6% B; 15–20 min at 6–11% B; 20–30 min at 11–11% B; 30–45 min at 11–20% B; 45–55 min at 20–30% B; 55–60 min at 30–90% B. The injection volume was 2 μL. DAD wavelength was set as 220 nm. Method validation was accorded to the Chinese Pharmacopeia guidelines, which were included linearity, limits of detection (LOD), limits of quantification (LOQ), repeatability, precision, stability and recovery.

### 3.5. UFLC/Q-TOF-MS Analysis

The identification was performed on a hybrid quadrupole time-of-flight tandem mass spectrometry (Exion-LC^TM^/X500R QTOF, AB SCIEX, Foster City, CA) equipped with an electrospray ionization (ESI) interface. The mass spectrometer was operated both in positive and negative ion mode. The parameters were set as following: the ion spray voltage of 5500/−4500 V; turbo spray temperature (TEM) of 550 °C; declustering potential (DP) of ±70 V; collision energy (CE) of ±40 V; nebulizer gas (gas 1) of 50 psi; heater gas (gas 2) of 50 psi and curtain gas of 30 psi [20,43]. Nitrogen was kept as the nebulizer and auxiliary gas. TOF/MS and TOF/MS-MS were scanned with the mass range of m/z 100–1200 and 50–1200, respectively. The experiments were run with 200 ms accumulation time for TOF/MS and 80 ms accumulation time for TOF/MS-MS. Continuous recalibration was carrying out at each 3 h. In addition, dynamic background subtraction (DBS) trigger information-dependent acquisition (IDA) was used to trigger acquisition of MS/MS of low-level constituents. The accurate mass and composition for the precursor ions and fragment ions were analyzed using the SCIEX OS software integrated with the instrument.

### 3.6. Antioxidant Activity Assay

After collection fractions, the antioxidant activity was tested for the 60 fractions by Total Antioxidant Capacity Assay Kit (ABTS). ABTS was oxidized to ABTS cation radical (ABTS•+) with appropriate oxidizing agent. The red wine has inhibited the production of ABTS•+ that can be detected by the absorbance at 734 nm. Initially, 0.4 mL ABTS stock solution was mixed with 0.4 mL of the oxidizing agent solution for the preparation of the radical. This solution was maintained at room temperature for 12–16 h in an amber bottle. Subsequently, the mixture was diluted in PBS to obtain an absorbance of approximately 0.7 ± 0.05. Finally, 200 μM ABTS was mixed with different fraction. The absorbance change at 734 nm was measured after 5 min. The reaction solution with PBS instead of test sample was used as a control test. The total antioxidant activity was expressed as Trolox-equivalent Antioxidant Capacity (TEAC) equivalent [51,52].

### 3.7. Thrombin Inhibitory Activity Assay

The inhibitory activity against thrombin was assessed by Thrombin Inhibitor Screening Kit (Fluorometric). Thrombin inhibitor screening principle was utilized the ability of thrombin to cleave a synthetic AMC-based peptide substrate to release AMC, which can be detected by measuring fluorescence intensity at Ex/Em = 350/450 nm. In the presence of thrombin inhibitors, the extent of cleavage reaction was reduced or completely abolished. The loss in fluorescence intensity can be correlated to the amount of inhibitor present in the assay solution. At first, the thrombin enzyme solution was prepared as 48 μL thrombin assay buffer and 2 μL thrombin enzyme stock solution in each well. Then, fractions were dissolved by methyl alcohol:water (1:1) and diluted to 10 times with thrombin assay buffer. The 10 μL diluted fractions or thrombin inhibitor control were added into the thrombin enzyme containing wells. The thrombin inhibitor control was consisted of 1 μL thrombin inhibitor and 9 μL thrombin assay buffer to thrombin enzyme well. The sample was incubated at room temperature for 15 min. Finally, the thrombin substrate solution was added into each well, which was consisted of 35 μL thrombin assay buffer and 5 μL thrombin substrate. Fluorescence was measured in a kinetic mode for 30–60 min at 37 °C (Ex/Em = 350/450 nm). The time points (T_1_ = 5 min, T_2_ = 10 min) were chosen in the linear range of the plot and obtain the corresponding values for the fluorescence (RFU1, RFU2). Irreversible inhibitors that inhibit the thrombin activity completely at the tested concentration will have ΔRFU = 0 and will show 100% relative inhibition. The inhibition (%) was calculated by (slope of enzyme control − slope of fraction)/slope of enzyme control × 100% [53].

### 3.8. Lipase Inhibitory Activity Assay

In this study, the inhibition of each fraction was assayed by measuring the fluorescence intensity of 4-methylumbelliferyl oleate (4-MU) (the hydrolytic product of 4-MUO). The assay was performed according to the method described with slight modifications. Initially, the concentrations of trypsin (0.83–30,000 U/mL) and time were investigated to establish linear range with the concentration of 4-MUO at 0.1 mM. Then, 25 μL fractions and 25 μL lipase solutions were mixed together and 50 μL 4-MUO was added to the mixture. Finally, the amount of 4-MU was measured by microplate reader at an excitation wavelength of 355 nm and an emission wavelength of 460 nm after incubating at 25 °C for 30 min. The inhibition of pancreatic lipase activity was calculated as follows: inhibition (%) = [(A_control_ − A_sample_)/A_control_ − (A_control_ − A_control sample_)/A_control_] × 100%, where A_sample_ was the fluorescence of the reactions with added fraction, A_control_ was the control, and A_control sample_ was the fluorescence of the solvent of the fraction [12,54].

### 3.9. Statistical Analyses

The results of precision and recovery have been measured in triplicates and the stability have been measured in sextuplicate, which were expressed as mean and RSD. The content of 12 compounds have been measured in triplicates and expressed as mean.

## 4. Conclusions

In this study, the multi-activity integrated strategy was proposed to screen, identify and quantify components from red wine for prevention of CHD via UHPLC-FC and UFLC-Q-TOF/MS. The multi-activity integrated strategy of red wine was established and verified the availability of screening the preventing CHD compounds from the complex sample. Red wine exhibited potential antioxidant activity and inhibitory activity against thrombin and lipase, which could be used to prevent CHD. In addition, the system had advantages over traditional methods in screening potential active compounds. The multi-activity integrated strategy may help to discover bioactive components rapidly and efficiently, which provided reference for exploring the health care function in food.

## Figures and Tables

**Figure 1 molecules-26-06750-f001:**
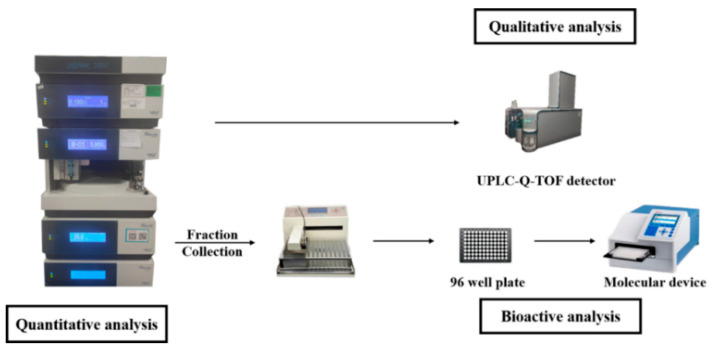
Schematic diagram of principle for screening active ingredient.

**Figure 2 molecules-26-06750-f002:**
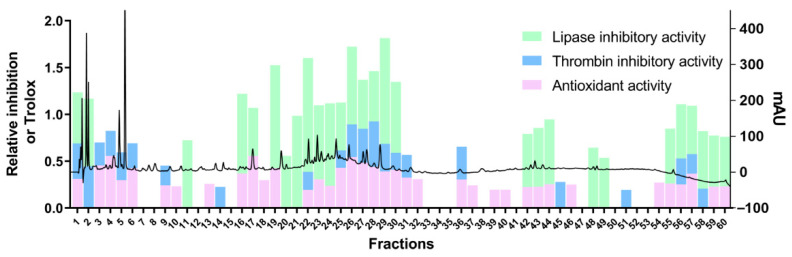
The multi-activity integrated chromatogram of antioxidant activity, inhibitory activity of thrombin and inhibitory activity of lipase. The chromatogram of red wine was shown in superstratum, substratum was a column diagram of each fraction for the three activities.

**Figure 3 molecules-26-06750-f003:**
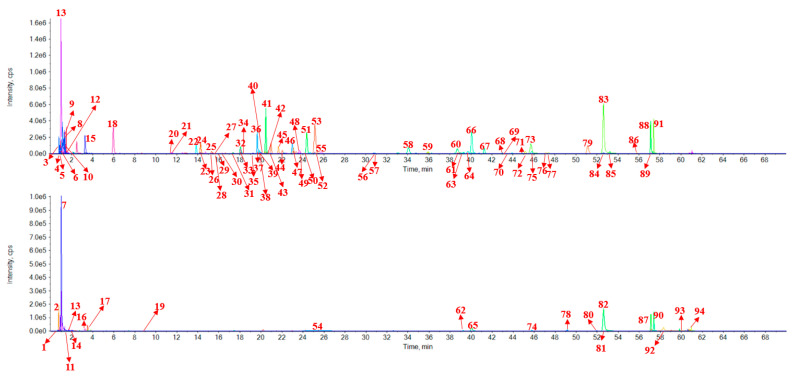
The extracted ion chromatogram of positive and negative ions.

**Figure 4 molecules-26-06750-f004:**
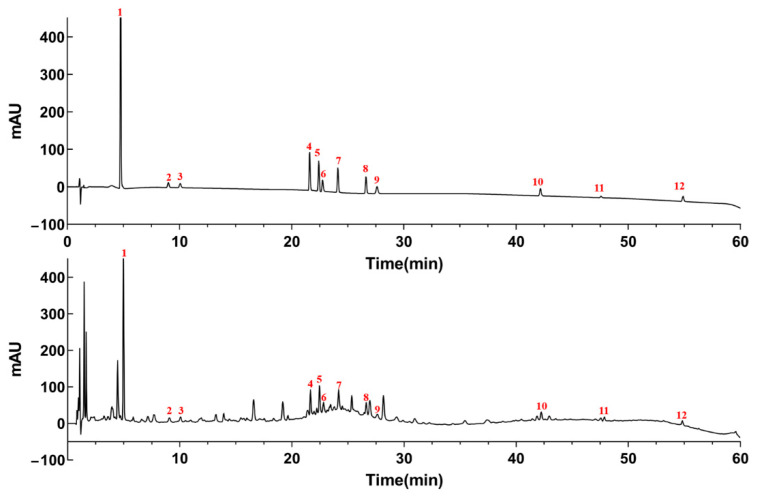
The typical reference chromatogram (top) and the typical chromatogram of red wine (bottom). gallic acid (1), coumalic acid (2), protocatechuic acid (3), proanthocyanidin B1 (4), catechins (5), caffeic acid (6), syringic acid (7), epicatechin (8), p-coumaric acid (9), isoquercitrin (10), isorhamnetin 3-O-glucoside (11) and quercetin (12).

**Figure 5 molecules-26-06750-f005:**
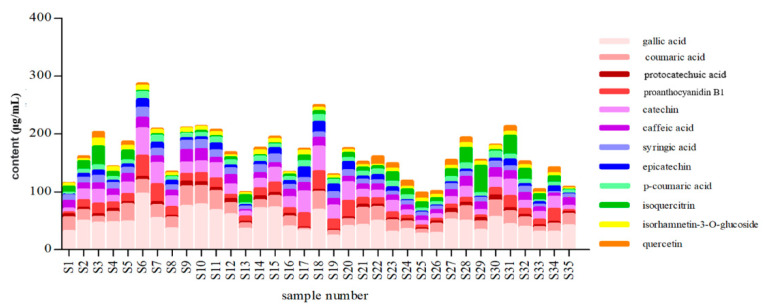
The content of 12 compounds from red wine (*n* = 3).

**Figure 6 molecules-26-06750-f006:**
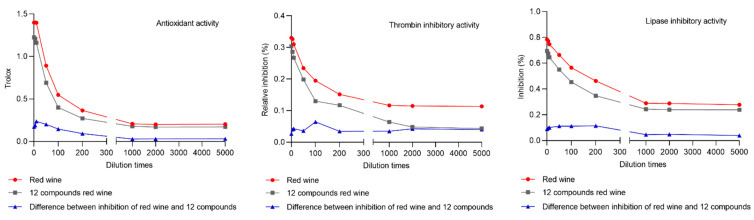
The inhibition ratios of red wine and 12 active compounds mixture with different dilution times and the difference between inhibition of red wine and 12 compounds mixture.

**Table 1 molecules-26-06750-t001:** The data of antioxidant activity (expressed in Trolox), thrombin inhibitory activity (expressed in inhibition) and lipase inhibitory activity (expressed in inhibition).

	Antioxidant Activity Inhibition (Trolox (%))	Thrombin Inhibitory Activity Inhibition (%)	Lipase Inhibitory Activity Inhibition (%)
F1	31.07	37.82	69.67
F2	no	44.42	87.30
F3	45.35	24.65	no
F4	55.64	26.68	no
F5	29.63	29.77	no
F6	42.56	26.61	no
F7	no	no	no
F8	no	no	no
F9	24.08	21.21	no
F10	23.10	no	no
F11	no	no	87.35
F12	no	no	no
F13	25.70	no	no
F14	no	22.44	no
F15	no	no	no
F16	36.79	no	100.20
F17	55.64	no	66.44
F18	29.63	no	no
F19	42.56	no	124.96
F20	no	no	70.68
F21	no	no	113.43
F22	19.34	19.22	136.53
F23	30.56	no	94.14
F24	23.59	no	103.15
F25	42.87	18.56	66.12
F26	54.32	35.06	97.89
F27	48.31	36.21	67.39
F28	43.88	48.47	68.79
F29	39.03	29.44	127.97
F30	40.83	18.07	90.74
F31	32.31	24.49	no
F32	30.82	no	no
F33	no	no	no
F34	no	no	no
F35	no	no	no
F36	30.36	35.12	no
F37	24.07	no	no
F38	no	no	no
F39	19.44	no	no
F40	19.48	no	no
F41	no	no	no
F42	22.28	no	71.92
F43	22.64	no	77.88
S44	25.36	no	84.26
F45	no	27.81	no
F46	25.08	no	no
F47	no	no	no
F48	no	no	79.38
F49	no	no	68.49
F50	no	no	no
F51	no	19.33	no
F52	no	no	no
F53	no	no	no
F54	27.09	no	no
F55	25.94	no	73.85
F56	25.27	27.66	72.88
F57	36.68	20.91	66.70
F58	no	20.65	76.42
F59	22.80	no	69.65
F60	23.21	no	67.84

**Table 2 molecules-26-06750-t002:** UFLC/Q-TOF-MS data and identification 94 compounds from red wine.

Peak No.	RT (min)	Ion	Found at Mass	MS/MS Fragmentation Ions	Formula	Identification	ppm
1	0.70	[M+H]^+^	116.0702	70.0647	C_5_H_9_NO_2_	D-proline	0.8
2	0.76	[M+H]^+^	175.1189	70.0652	C_6_H_14_N_4_O_2_	L-arginine	−0.3
3	0.79	[M−H]^−^	179.0562	99.0087, 71.0137, 59.0139	C_6_H_12_O_6_	D-glucose	0.3
4	0.88	[M-H]^−^	181.0720	83.0143, 71.0140, 59.0140	C_6_H_14_O_6_	D-mannitol	1.5
5	0.96	[M−H]^−^	191.0199	85.0296, 59.0134	C_6_H_8_O_7_	citric acid	1.0
6	0.99	[M−H]^−^	149.0091	105.0217, 87.0085, 72.9928	C_4_H_6_O_6_	tartaric acid	−0.7
7	1.04	[M+H]^+^	138.0548	120.0433, 92.0495, 78.0339	C_7_H_7_NO_2_	4-aminobenzoic acid	−1.5
8	1.12	[M−H]^−^	341.1088	161.0459, 71.0137, 59.0138	C_12_H_22_O_11_	trehalose	−0.3
9	1.15	[M−H]^−^	149.0455	75.0221, 59.0133	C_5_H_10_O_5_	D-ribose	−0.2
10	1.31	[M−H]^−^	133.0142	115.0018, 71.0136	C_4_H_6_O_5_	L-malic acid	−0.5
11	1.33	[M+H]^+^	118.0862	72.0809	C_5_H_11_NO_2_	L-valine	−0.4
12	1.48	[M−H]^−^	173.0454	137.0225, 93.0343	C_7_H_10_O_5_	shikimic acid	−1.0
13	1.71	[M+H]^+^	150.0586	74.0243, 61.0108	C_5_H_11_NO_2_S	L-methionine	1.8
14	2.03	[M+H]^+^	124.0393	80.0506, 55.9346	C_6_H_5_NO_2_	niacin	0.2
15	3.27	[M−H]^−^	117.0192	99.9254, 73.0291	C_4_H_6_O_4_	succinic acid	−1.2
16	3.51	[M+H]^+^	165.0547	119.0479, 95.0503, 77.0388, 59.9302	C_9_H_8_O_3_	trans-4-hydroxycinnamic acid	0.7
17	3.85	[M+H]^+^	137.0913	121.0616, 103.0616, 95.0561, 91.0556, 77.0391	C_8_H_11_NO	tyramine	−0.5
18	5.97	[M−H]^−^	169.0141	125.0244, 107.0156, 79.0186	C_7_H_6_O_5_	gallic acid	−1.0
19	8.77	[M+H]^+^	141.0182	123.0080, 95.0127, 67.0187, 55.9453	C_6_H_4_O_4_	coumalic acid	−0.1
20	11.49	[M−H]^−^	153.0558	123.0085, 108.0221, 95.0139, 78.9590	C_8_H_10_O_3_	hydroxytyrosol	0.2
21	11.57	[M−H]^−^	153.0194	109.0298, 91.0298, 65.0029	C_7_H_6_O_4_	protocatechuic acid	0.3
22	13.85	[M−H]^−^	305.0664	179.0360, 165.0194, 137.0243, 125.0246	C_15_H_14_O_7_	(+)-gallocatechin	−1.0
23	14.27	[M−H]^−^	179.0349	135.0560, 89.0394, 79.0550	C_9_H_8_O_4_	acetylsalicylic acid	−0.7
24	14.27	[M−H]^−^	179.0348	161.0263, 135.0454, 117.0351	C_13_H_12_O_9_	caftaric acid	−1.0
25	15.05	[M−H]^−^	153.0194	123.0061, 109.0298, 91.0298	C_7_H_6_O_4_	gentisic acid	0.4
26	15.21	[M−H]^−^	137.0255	108.0213	C_7_H_6_O_3_	protocatechualdehyde	−2.7
27	15.31	[M−H]^−^	181.0506	135.0453	C_9_H_10_O_4_	homovanillic acid	0.0
28	15.43	[M−H]^−^	183.0299	124.0234, 95.0136	C_8_H_8_O_5_	methyl gallate	0.0
29	15.90	[M−H]^−^	137.0244	93.0340	C_7_H_6_O_3_	4-hydroxybenzoic acid	0.4
30	16.32	[M−H]^−^	353.0877	191.0616, 179.0341, 135.0439	C_16_H_18_O_9_	chlorogenic acid	−0.2
31	17.36	[M−H]^−^	295.0458	163.0400, 119.0503, 87.0094	C_13_H_12_O_8_	p-coutaric acid	0.1
32	18.08	[M−H]^−^	163.0401	119.0498, 93.0347	C_9_H_8_O_3_	m-coumaric acid	0.2
33	18.08	[M−H]^−^	295.0458	163.0400, 119.0498	C_13_H_12_O_8_	coutaric acid	−0.4
34	18.15	[M−H]^−^	145.0507	73.0273, 55.0191	C_6_H_10_O_4_	dimethyl succinate	0.6
35	19.24	[M−H]^−^	121.0296	108.0212, 92.0265	C_7_H_6_O_2_	benzoic acid	0.8
36	19.66	[M−H]^−^	577.1346	425.0864, 407.0775, 289.0714, 161.0238, 125.0243	C_30_H_26_O_12_	procyanidin B1	−0.9
37	19.89	[M−H]^−^	305.0664	245.0462, 219.0652, 139.0401, 125.0244	C_15_H_14_O_7_	(−)-epigallocatechin	0.2
38	19.98	[M−H]^−^	577.1346	407.0768, 289.0709	C_30_H_26_O_12_	procyanidin B3	0.0
39	20.01	[M−H]^−^	167.0350	108.0212	C_8_H_8_O_4_	vanillic acid	0.2
40	20.21	[M−H]^−^	163.0400	119.0502, 93.0343	C_9_H_8_O_3_	2-Hydroxycinnamic acid	−0.1
41	20.47	[M−H]^−^	289.0713	245.0843, 203.0713, 151.0401, 109.0294	C_15_H_14_O_6_	catechin	−1.5
42	20.58	[M−H]^−^	325.0563	193.0490, 117.0341	C_14_H_14_O_9_	fertaric acid	−0.5
43	20.98	[M−H]^−^	179.0349	135.0449, 89.0393	C_9_H_8_O_4_	caffeic acid	−1.0
44	21.90	[M−H]^−^	441.0186	289.0735, 217.0167, 191.0337, 151.0386	C_22_H_18_O_10_	epicatechin gallate	2.2
45	22.06	[M−H]^−^	197.0454	182.0256, 167.0014, 138.0317, 123.0090, 95.0142	C_9_H_10_O_5_	syringic acid	−0.3
46	23.01	[M−H]^−^	577.1346	425.0893, 407.0782, 289.0721	C_30_H_26_O_12_	procyanidin B2	−0.7
47	23.11	[M−H]^−^	325.0926	163.0394, 145.0294, 117.0343	C_15_H_18_O_8_	p-coumaric acid glucoside	−0.5
48	23.37	[M−H]^−^	189.0769	71.0503	C_8_H_14_O_5_	diethyl malate	0.3
49	23.69	[M−H]^−^	165.0556	119.0527, 103.0565	C_9_H_10_O_3_	desaminotyrosine	−0.5
50	24.37	[M−H]^−^	137.0244	93.0470, 65.0473	C_7_H_6_O_3_	2-hydroxybenzoic acid	2.2
51	24.38	[M−H]^−^	289.0713	203.0710, 187.0396, 151.0401, 123.0449, 109.0290	C_15_H_14_O_6_	epicatechin	−0.7
52	24.68	[M−H]^−^	167.0350	123.0436, 108.0212	C_8_H_8_O_4_	3-hydroxy-4-methoxybenzoic acid	2.5
53	25.16	[M−H]^−^	197.0454	169.0142, 125.0239, 79.0184	C_9_H_10_O_5_	ethyl gallate	−0.9
54	25.32	[M]^+^	493.1341	331.0822, 316.0553	C_23_H_25_O_12_	malvidin-3-O-glucoside	0.1
55	25.74	[M−H]^−^	163.0401	119.0503, 93.0347	C_9_H_8_O_3_	p-coumaric acid	−0.6
56	30.75	[M−H]^−^	167.0350	123.0450, 65.0024	C_8_H_8_O_4_	homogentisic acid	0.8
57	30.80	[M−H]^−^	577.1346	425.0886, 407.0789, 289.0734	C_30_H_26_O_12_	procyanidin B7	0.6
58	34.06	[M−H]^−^	479.0826	316.0225, 271.0249, 151.0034	C_21_H_20_O_13_	myricetin-3-O-galactoside	−1.0
59	35.93	[M−H]^−^	389.1242	227.0716, 185.0606, 143.0505	C_20_H_22_O_8_	trans-piceid	−0.2
60	38.70	[M−H]^−^	300.9987	257.011, 229.0126, 145.0295	C_14_H_6_O_8_	ellagic acid	−0.9
61	38.84	[M−H]^−^	181.0506	109.0286, 65.0032	C_9_H_10_O_4_	dihydrocaffeic acid	0.5
62	39.28	[M]^+^	465.1036	303.051, 229.0488	C_21_H_21_O_12_	delphinidin-3-O-glucoside	1.9
63	39.31	[M−H]^−^	463.0884	301.0379, 271.0245, 243.0291, 151.0031	C_21_H_20_O_12_	hyperoside	0.4
64	39.71	[M−H]^−^	449.1090	285.0413, 151.0041	C_21_H_22_O_11_	quercitrin	0.2
65	39.83	[M]^+^	627.1462	303.0351	C_27_H_31_O_17_	delphinidin-3,5-O-diglucoside	1.1
66	40.08	[M−H]^−^	477.0670	301.0352, 255.0296, 151.0034	C_21_H_18_O_13_	quercetin-3-O-glucuronide	−0.9
67	41.28	[M−H]^−^	493.0985	330.0381, 315.0155, 271.0249, 151.0034	C_22_H_22_O_13_	laricitrin-3-O-glucoside	−0.6
68	42.86	[M−H]^−^	447.0936	301.0372, 271.0256, 255.0295, 151.0020	C_21_H_20_O_11_	luteolin -3’-glucoside	1.2
69	42.98	[M−H]^−^	449.1090	285.0413, 151.0041	C_21_H_22_O_11_	astilbin	0.9
70	43.45	[M−H]^−^	463.0884	301.0379, 255.0305, 151.0039	C_21_H_20_O_12_	isoquercitrin	0.8
71	44.78	[M−H]^−^	447.0936	301.0372, 255.0293, 151.0016	C_21_H_20_O_11_	astragalin	0.8
72	45.19	[M−H]^−^	389.1242	227.0994, 185.0606	C_20_H_22_O_8_	cis-piceid	0.0
73	45.72	[M−H]^−^	317.0300	245.0504, 137.0241, 109.0292, 151.0035	C_15_H_10_O_8_	myricetin	−0.9
74	45.77	[M]^+^	479.1193	317.0648, 302.0429, 285.0362, 153.0184	C_22_H_23_O_12_	petunidin-3-O-glucoside	1.9
75	45.80	[M−H]^−^	477.1041	315.0530, 300.0263, 271.0255, 243.0306, 151.0036	C_22_H_22_O_12_	isorhamnetin-3-O-glucoside	0.6
76	47.17	[M−H]^−^	227.0715	185.0587, 143.0501, 115.0556	C_14_H_12_O_3_	trans-resveratrol	0.6
77	47.42	[M−H]^−^	435.1300	273.0769, 179.0349, 167.0352	C_21_H_24_O_10_	phloridzin	0.7
78	49.16	[M]^+^	507.1141	303.0767, 187.0626, 113.0234	C_23_H_23_O_13_	delphindin-3-O-(6-O-acetylglucoside)	1.6
79	51.13	[M−H]^−^	207.0662	179.0433, 135.0446	C_11_H_12_O_4_	ethyl caffeate	−0.6
80	51.94	[M]^+^	639.1719	477.1176, 419.0853, 331.0874	C_32_H_31_O_14_	malvidin-3-O-(6-O-coumaroylglucoside)	1.7
81	52.60	[M+H]^+^	303.0497	285.0398, 257.0456, 229.0492, 153.0188	C_15_H_10_O_7_	morin	−0.8
82	52.60	[M]^+^	303.0497	229.0492, 153.0188	C_15_H_11_O_7_	delphinidin	−0.8
83	52.63	[M−H]^−^	301.0351	245.0447, 229.0513, 151.0034, 121.0292	C_15_H_10_O_7_	quercetin	−1.0
84	52.67	[M−H]^−^	227.0715	143.0504	C_14_H_12_O_3_	resveratrol	−0.9
85	53.23	[M−H]^−^	331.0459	316.0231, 271.0251, 178.9991, 151.0039	C_16_H_12_O_8_	laricitrin	0.0
86	55.82	[M−H]^−^	271.0613	229.0500, 177.0190, 151.0048, 119.0618	C_15_H_12_O_5_	naringenin	0.4
87	57.10	[M]^+^	287.0549	213.0559, 153.0180, 137.0233	C_15_H_11_O_6_	cyanidin	−0.3
88	57.12	[M−H]^−^	285.0403	239.0356, 187.0403, 151.004	C_15_H_10_O_6_	kamepferol	−0.6
89	57.32	[M−H]^−^	345.0618	330.0385, 315.0153, 259.0252, 187.0394	C_17_H_14_O_8_	syringetin	0.6
90	57.39	[M]^+^	317.0656	229.0498, 153.0188	C_16_H_13_O_7_	petunidin	0.0
91	57.41	[M−H]^−^	315.0508	300.0270, 271.0241, 255.0299, 227.0241, 151.0036	C_16_H_12_O_7_	isorhamnetin	−0.6
92	58.32	[M+H]^+^	193.1588	56.9419	C_13_H_20_O	ionone	0.6
93	60.04	[M+H]^+^	173.1536	76.0221	C_10_H_20_O_2_	ethyl caprylate	0.2
94	60.96	[M+H]^+^	195.1748	177.1634, 107.0855, 77.0387	C_13_H_22_O	theaspirane	2.1

**Table 3 molecules-26-06750-t003:** The corresponding relationship between fraction number and peak number of the 77 potential active components.

Fraction Number	Peak Number	Identification	Antioxidant Activity	Thrombin Inhibitory Activity	Lipase Inhibitory Activity
F1	5, 6, 10, 12	citric acid, tartaric acid, L-malic acid, shikimic acid	yes	yes	yes
F2	12	shikimic acid		yes	yes
F3	15, 16	succinic acid, trans-4-hydroxycinnamic acid	yes	yes	
F4	17	tyramine	yes	yes	
F5	18	gallic acid	yes	yes	
F6	18	gallic acid	yes	yes	
F9	19	coumalic acid	yes	yes	
F10	21	protocatechuic acid	yes		
F11	21	protocatechuic acid			yes
F13	22	(+)-gallocatechin	yes		
F14	23, 24	acetylsalicylic acid, caftaric acid		yes	
F16	25, 26, 27, 28	gentisic acid, protocatechualdehyde, homovanillic acid, methyl gallate	yes		yes
F17	29	4-hydroxybenzoic acid	yes		yes
F18	30	chlorogenic acid	yes		
F19	31	p-coutaric acid	yes		yes
F20	32, 33, 34	m-coumaric acid, coutaric acid, dimethyl succinate			yes
F21	35	benzoic acid			yes
F22	36, 37, 38	procyanidin B1, (−)-epigallocatechin, procyanidin B3	yes	yes	yes
F23	39, 40, 41, 42, 43	vanillic acid, 2-Hydroxycinnamic acid, catechin, fertaric acid, caffeic acid	yes		yes
F24	44	epicatechin gallate	yes		yes
F25	45	syringic acid	yes	yes	yes
F26	46, 47, 48	procyanidin B2, p-coumaric acid glucoside, diethyl malate	yes	yes	yes
F27	49, 50, 51	desaminotyrosine, 2-hydroxybenzoic acid, epicatechin	yes	yes	yes
F28	53, 54	ethyl gallate, malvidin-3-O-glucoside	yes	yes	yes
F29	55	p-coumaric acid	yes	yes	yes
F30	55	p-coumaric acid	yes	yes	yes
F31	56	homogentisic acid	yes	yes	
F32	57	procyanidin B7	yes		
F36	58	myricetin-3-O-galactoside	yes	yes	
F37	59	trans-piceid	yes		
F39	60, 61	ellagic acid, dihydrocaffeic acid	yes		
F40	62, 63	delphinidin-3-O-glucoside, hyperoside	yes		
F42	65, 66	delphinidin-3,5-O-diglucoside, quercetin-3-O-glucuronide	yes		yes
F43	67, 68, 69, 70	laricitrin-3-O-glucoside, luteolin -3-O-glucoside, astilbin, isoquercitrin	yes		yes
F44	70	isoquercitrin	yes		yes
F45	71	astragalin		yes	
F46	72	cis-piceid	yes		
F48	73, 74, 75	myricetin, petunidin-3-O-glucoside, isorhamnetin-3-O-glucoside			yes
F49	76, 77	trans-resveratrol, phloridzin			yes
F51	778	delphindin-3-O-(6-O-acetylglucoside)		yes	
F54	79, 80	ethyl caffeate, malvidin-3-O-(6-O-coumaroylglucoside)	yes		
F55	81, 82, 83, 84	morin, delphinidin, quercetin, resveratrol	yes		yes
F56	85	laricitrin	yes	yes	yes
F57	85	laricitrin	yes	yes	yes
F58	86	naringenin		yes	yes
F59	87	cyanidin	yes		yes
F60	88, 89, 90	kamepferol, syringetin, petunidin	yes		yes

**Table 4 molecules-26-06750-t004:** Precision, stability, linearity, LODs and LOQs of 12 compounds from red wine.

Analytes	Precision				Stability	Linearity			LOQ (μg/mL)	LOD (μg/mL)
	Intra-Day, RSD (%), (*n* = 3)			Inter-Day RSD (%) (*n* = 9)	RSD (%) (*n* = 6)	Range (μg/mL)	Equation	r^2^		
	Low	Middle	High							
gallic acid	2.2	2.2	1.3	3.4	2.5	20.00~320.0	y = 0.3627x + 0.0831	0.9998	0.12	0.040
coumalic acid	3.2	1.5	1.9	4.4	1.8	6.030~96.48	y = 0.2339x − 0.0078	0.9994	2.0	0.75
protocatechuic acid	4.9	2.2	1.7	4.2	1.1	1.005~16.08	y = 0.1498x + 0.0290	0.9993	0.12	0.040
proanthocyanidin B1	3.6	2.1	1.8	4.6	2.2	6.000~96.0	y = 0.1156x + 0.0121	0.9996	0.50	0.12
catechin	5.0	1.3	2.0	3.3	2.5	6.030~96.5	y = 0.2149x + 0.0288	0.9997	0.12	0.040
caffeic acid	2.5	2.3	1.3	4.7	2.2	3.000~48.00	y = 0.2645x + 0.0428	0.9992	0.12	0.040
syringic acid	4.8	2.4	2.1	4.8	1.8	2.973~47.56	y = 0.5234x + 0.0300	0.9997	0.25	0.08
epicatechin	4.2	2.0	1.2	4.4	2.9	3.030~48.48	y = 0.3627x + 0.0831	0.9998	0.50	0.12
p-coumaric acid	4.4	1.7	1.4	4.1	2.1	3.015~48.24	y = 0.2339x + 0.0078	0.9994	0.25	0.08
isoquercitrin	4.6	1.9	1.3	4.6	1.8	0.2500~64.64	y = 0.1495x + 0.0431	0.9994	0.25	0.08
isorhamnetin-3-O-glucoside	5.0	2.1	1.7	3.3	2.3	0.990~15.84	y = 0.1156x + 0.0121	0.9996	0.50	0.12
quercetin	4.3	2.6	2.2	4.6	2.5	0.7500~31.46	y = 0.2166x − 0.0094	0.9995	0.75	0.25

**Table 5 molecules-26-06750-t005:** The recovery of 12 compounds from red wine (*n* = 3).

Analytes	Original (μg)	Spiked (μg)	Found (μg)	Recovery (%)	RSD (%)
gallic acid	29.34	20.00	45.81	99.8	1.8
		40.00	68.85	97.0	1.4
		60.00	89.14	99.8	1.0
coumalic acid	9.724	6.030	14.65	87.8	4.1
		12.06	21.19	95.1	2.2
		18.09	26.77	86.7	3.1
protocatechuic acid	3.298	1.005	3.720	83.4	4.3
		2.010	4.864	90.6	2.5
		3.015	6.338	98.4	1.4
proanthocyanidin B1	9.027	6.000	14.01	89.7	4.0
		12.00	21.58	103.6	2.3
		18.00	26.27	94.1	2.1
catechin	11.43	3.000	14.15	97.7	3.6
		6.000	17.50	100.1	0.7
		9.000	20.61	100.0	0.3
caffeic acid	6.539	2.972	9.33	96.3	3.4
		5.945	13.14	105.0	2.1
		8.918	15.26	98.9	1.0
syringic acid	6.204	2.973	8.97	97.7	3.4
		5.945	12.08	97.5	3.2
		8.918	14.95	101.7	1.8
epicatechin	3.922	3030	6.932	101.2	1.6
		6.060	10.04	101.9	0.7
		9.090	13.18	105.9	1.4
p-coumaric acid	4.473	3.015	7.590	105.6	3.6
		6.030	10.15	92.5	1.1
		9.045	13.78	100.6	0.6
isoquercitrin	8.802	4.040	12.75	98.1	3.2
		8.080	16.75	98.8	2.1
		12.12	20.64	98.8	1.9
isorhamnetin-3-O-glucoside	3.144	0.9900	4.053	98.1	1.1
		1.980	5.123	97.4	2.6
		2.970	6.019	102.1	2.4
quercetin	3.146	1.966	4.928	95.5	1.6
		3.933	6.950	96.7	1.3
		5.899	8.82	92.2	4.1

## Data Availability

The MS spectrums of all compounds were shown in Appendix A.

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
