# Peer review of "Based on Multi-Activity Integrated Strategy to Screening, Characterization and Quantification of Bioactive Compounds from Red Wine"

_molecules, 2021, doi:10.3390/molecules26216750_

Round 1

Reviewer 1 Report

The authors performed a screening using chromatographic techniques to analyze bioactive compounds.   The manuscript is flawed and cannot be considered for publication.   General comments 1) The introduction needs to be further explored with a systematic review of the state of the art. 2) Authors do not need to add the molecular formulas of the identified substances. 3) The discussion is confusing and needs improvement. 4) Materials and methods do not have references, this is a serious error, as there is no support for other authors to evaluate the methods if they are correct. Authors must remember that materials and methods are very important in an MS.

Author Response

To Reviewer 1:

Reviewer #1: The authors performed a screening using chromatographic techniques to analyze bioactive compounds. The manuscript is flawed and cannot be considered for publication.

The introduction needs to be further explored with a systematic review of the state of the art.

Response: Thank you very much for your suggestion. In order to further explore with a systematic review, we have supplemented the traditional method for screening active compounds in Page 2, Line 54 and the overview of the HPLC methods that have been used to separate compounds in Page 2, Line 62.

Authors do not need to add the molecular formulas of the identified substances. Response: The molecular formula has removed.

The discussion is confusing and needs improvement.

Response: Thank you for your suggestion. In order to improve discussion section, we have supplemented the discussion clearly in Page 3, Line 101 for establishment of multi-activity integrated strategy, active ingredients of red wine in Page 11, Line 272 and the application of multi-activity integrated strategy in Page 16, Line 364. Red wine has many ingredients, different compounds have different activities, such as antioxidant, anticoagulant and lipid-lowering. CHD were the result of the interaction of many complex factors. The multi-activity of red wine was corresponding to the multiple pathogenesis of CHD. Indeed, the multi-activity integrated strategy of red wine was established to screen 12 bioactive compounds for quality evaluation to prevent CHD. Additionally, the multi-activity integrated strategy may help to discover bioactive components rapidly and efficiently, which provided reference for exploring active ingredients in food.

Materials and methods do not have references, this is a serious error, as there is no support for other authors to evaluate the methods if they are correct. Authors must remember that materials and methods are very important in an MS.

Response: Thank you very much for your useful advice. We have supplemented references to the materials and methods in Page 17, Line 411, 412, 432 and Page 18, Line 446, 468, 482.

Reviewer 2 Report

In this article, the authors describe the UHPLC-FC separation of components in red wine, their identification by UFLC-Q-TOF/MS, and the quantification of twelve of them. They also investigate the antioxidant activity, thrombin inhibitory activity, and lipase inhibitory activity of the collected fractions.

In the introduction, I completely miss an overview of the HPLC methods that have been used to separate compounds in wine. Why did the authors use this HPLC column for the separation? On what were the authors based? 

Table 2: In Chapter 3, the authors state that the results of antioxidant activity, thrombin inhibitory activity, and lipase inhibitory activity are expressed as TEAC equivalent, % relative inhibition, and % inhibition, respectively. Thus, these values should be reported in the table. There is only "yes" or nothing. Some values can be read from the bar chart in Fig. 1, but this is only a guide.

Fig. 1: I do not understand what an " multi-activity integrated chromatogram" is. There is no more detailed description.

Fig. 3: Why are there two chromatograms? What is the difference between them?

There is no discussion.

Author Response

To Reviewer 2:

Reviewer #2: In this article, the authors describe the UHPLC-FC separation of components in red wine, their identification by UFLC-Q-TOF/MS, and the quantification of twelve of them. They also investigate the antioxidant activity, thrombin inhibitory activity, and lipase inhibitory activity of the collected fractions.

In the introduction, I completely miss an overview of the HPLC methods that have been used to separate compounds in wine. Why did the authors use this HPLC column for the separation? On what were the authors based?

Response: Thank you very much for your question. In order to enrich the introduction to overview the HPLC methods, we have supplemented the traditional method for screening active compounds and the overview of the HPLC methods that have been used to separate compounds in wine in Page 2, Line 62. The HPLC column (HSS T3, 1.8 μm, 2.1 × 100 mm, Waters, Ireland) can withstand 100% water mobile phase, and the particle size was only 1.8 microns, which was adapted for the separation of complex samples especially for the strong polar compounds in red wine.

In Chapter 3, the authors state that the results of antioxidant activity, thrombin inhibitory activity, and lipase inhibitory activity are expressed as TEAC equivalent, % relative inhibition, and % inhibition, respectively. Thus, these values should be reported in the table. There is only "yes" or nothing. Some values can be read from the bar chart in Fig. 1, but this is only a guide.

Response: Thank you very much for your suggestion. We have supplemented Table 1. for the detail results of antioxidant activity, thrombin inhibitory activity, and lipase inhibitory activity in Page 3, Line 110.

I do not understand what an " multi-activity integrated chromatogram" is. There is no more detailed description.

Response: The figure caption will be changed to “The multi-activity integrated chromatogram of antioxidant activity, inhibitory activity of thrombin and inhibitory activity of lipase: The chromatogram of red wine was shown superstratum, substratum is a column diagram of each fraction for the three activities.” in Page 3, Line 106.

Fig. 3: Why are there two chromatograms? What is the difference between them?

Response: Thank you for your advice. The figure caption will be changed to “The above image for reference typical chromatogram and the below image for red wine typical chromatogram. gallic acid (1), coumalic acid (2), protocatechuic acid (3), proanthocyanidin B1 (4), catechins (5), caffeic acid (6), syringic acid (7), epicatechin (8), p-coumaric acid (9), isoquercitrin (10), isorhamnetin-3-O-glucoside (11) and quercetin (12).” in Page 13, Line 313.

Reviewer 3 Report

Type of the Paper (Article)

Title: Based on multi-activity integrated strategy to screening, characterization and quantification the bioactive compounds of red wine

This work is well presented:

Minor error:

Comment. UHPLC-PDA-FC: PDA acronym is not mention before paragraph 69. Full definition of UHPLC-PDA-FC in paragraph 69.

Author Response

To Reviewer 3:

Reviewer #3: This work is well presented.

UHPLC-PDA-FC: PDA acronym is not mention before paragraph 69. Full definition of UHPLC-PDA-FC in paragraph 69.

Response: Thank you for your correction. The full definition of P UHPLC-PDA-FC has revised to ultra high performance liquid chromatography-photo-diode array-fraction collector.

Reviewer 4 Report

The manuscript is interesting and may have novelty, however language expression should be improved throughout the manuscript. Following major concerns should be addressed before consideration.

Title: Please correct the title as "quantification of bioactive compounds from red wine.

Abstract: Change the context and language expression for the first line of of abstract.

The results of bioactivity assays should be highlighted with numerical terms in abstract. 

Language expression is required to be improved. 

Introduction: 

Accumulated studies??

The background information is not sufficient, authors should highlight the previous research studies based on screening of bioactive compounds in wine and identification of potential biological benefits. 

Results:

In abstract 87 Active compounds were detected however, in results 108 active components have been mentioned?

Figure 2 is not clear.

Table 2: Reporting the activity (Antioxidant, thrombin and lipase inhibitory activity) as yes or no is not acceptable. authors should have support the data with concrete values obtained for each assay.

What was the reason of selecting 12 particular compounds for quantification by UHPLC?

There is no evidence of statistical analysis.

Methods: 

3.6, 3.7 and 3.8 references should be mentioned for each bioactivity assay.

Discussion:

There is negligible discussion, each section of result should be supported by discussions. 

Author Response

To Reviewer 4:

Reviewer #4: The manuscript is interesting and may have novelty, however language expression should be improved throughout the manuscript. Following major concerns should be addressed before consideration.

Title: Please correct the title as quantification of bioactive compounds from red wine.

Response: Thank you very much for your useful advice. The title has revised to “Based on multi-activity integrated strategy to screening, characterization and quantification of bioactive compounds from red wine.”

Change the context and language expression for the first line of abstract.

Response: The context and language expression in the first line of abstract has revised to “According to French Paradox, red wine was famous for the potential effects on coronary heart disease (CHD), but the specific compounds of the effect was unclear. Therefore, screening and characterization of bioactive compounds from red wine was extremely necessary.” in Page 1, Line 11.

The results of bioactivity assays should be highlighted with numerical terms in abstract.

Response: We have supplemented the numerical terms of antioxidant activity, thrombin inhibitory activity, and lipase inhibitory activity in Page 1, Line 20.

Language expression is required to be improved.

Response: The language expression has amended carefully.

Introduction:

Accumulated studies??

The background information is not sufficient, authors should highlight the previous research studies based on screening of bioactive compounds in wine and identification of potential biological benefits.

Response: Thank you for your advice. We have supplemented the previous research studies for screening active compounds in Page 2, Line 54. The traditional method for screening active compounds was extracted the single component from the sample and evaluated the biological activity by pharmacological means. The animal model was the most important screening models in the part of bioactive components evaluation, which can clearly response to the efficacy of ingredients in the whole sample. However, the method was high cost and low efficiency. Be-sides, compared with the animal models, cellular models were relatively cheap and convenient but the it has higher required on instrumentation.

Results:

In abstract 87 Active compounds were detected however, in results 108 active components have been mentioned? What was the reason of selecting 12 particular compounds for quantification by UHPLC?

There is no evidence of statistical analysis.

Response: The five ensemble approaches were applicated for selecting active ingredients to the quality evaluation of red wine. The analytical data of active fractions with antioxidant activity, inhibitory activity of thrombin or lipase and 106 compounds of mass spectrometry were correlated according to retention time and peak order. 87 compounds were screened for the prevention and treatment of CHD. 63 compounds had multiple activities; 24 compounds had single activity, and 16 compounds had a strong single activity, which were protocatechuic acid, guaiacol, (+)-catechin gallate, (+)-epigallocatechin, aspirin, caftaric acid, chlorogenic acid, benzoic acid, procyanidin B7, caffeoylshikimic acid, epicatechin gallate, trans-piceid, astragalin, cis-piceid, ethyl caffeate and malvi-din-3-O-(6-O-coumaroylglucoside). Therefore, 79 ingredients had multiple activities or had a strong single activity. According to the reference substance was easy to obtain and the activity of this ingredient has been reported many times in literature, a total of 21 compounds were screened, which were succinic acid, gallic acid, coumalic acid, procya-nidin B1, (-)-epigallocatechin, vanillic acid, catechin, caffeic acid, syringic acid, epicate-chin, p-coumaric acid, salicylic acid, quercetin-3-O-glucuronide, isoquercitrin, isorham-netin-3-O-glucoside, trans-resveratrol, quercetin, protocatechuic acid, (+)-catechin gallate, (+)-epigallocatechin and epicatechin gallate. The statistical analysis in Page 11, Line 284. The statistical analysis was added in Page 18, Line 483.

Figure 2 is not clear.

Response: Figure 2 was added the number of peaks in Page 10, Line 269.

Table 2: Reporting the activity (Antioxidant, thrombin and lipase inhibitory activity) as yes or no is not acceptable. authors should have support the data with concrete values obtained for each assay.

Response: We have supplemented Table 1. for the results of antioxidant activity, thrombin inhibitory activity, and lipase inhibitory activity in Page 3, Line 106.

Methods:

3.6, 3.7 and 3.8 references should be mentioned for each bioactivity assay.

Response: We have supplemented references to antioxidant, thrombin and lipase inhibitory activity assay in Page 18, Line 446, 468 and 482.

Discussion:

There is negligible discussion, each section of result should be supported by discussions.

Response: Thank you for your advice. We have supplemented the discussion clearly in Page 3, Line 101 for establishment of multi-activity integrated strategy, active ingredients of red wine in Page 10, Line 276 and the application of multi-activity integrated strategy in Page 16, Line 364. Red wine has many ingredients, different compounds have different activities, such as antioxidant, anticoagulant and lipid-lowering. CHD were the result of the interaction of many complex factors. The multi-activity of red wine was corresponding to the multiple pathogenesis of CHD. Indeed, the multi-activity integrated strategy of red wine was established to screen 12 bioactive compounds for quality evaluation to prevent CHD. Additionally, the multi-activity integrated strategy may help to discover bioactive components rapidly and efficiently, which provided reference for exploring active ingredients in food.

Round 2

Reviewer 1 Report

The authors complied with all the recommendations I recommend the manuscript for publication

Author Response

Response: Thank you very much for your advice.

Reviewer 2 Report

I am satisfied with the corrections in the article, but editing of English is still needed.

I would change the description of Fig. 4 as follows: The typical reference chromatogram (top) and the typical chromatogram of red wine (bottom).

Author Response

Response: Thank you very much for your correction. The description of Figure 4 has revised to “the typical reference chromatogram (top) and the typical chromatogram of red wine (bottom)” in Page 13, Line 313.

Reviewer 4 Report

The manuscript has been revised, however authors should mentioned experiments particularly Table 1 were performed in triplicates or not? If yes mentioned mean and SD. Still statistical analysis is not accessible anywhere in the study. The language expression can be improved for clarity of the information presented

Author Response

The manuscript has been revised, however authors should mentioned experiments particularly Table 1 were performed in triplicates or not? If yes mentioned mean and SD.

Response: Thank you very much for your useful advice. The experiments of Table 1 were performed just once.

Still statistical analysis is not accessible anywhere in the study.

Response: Statistical analysis was supplemented in Page 18, Line 487. The results of precision and recovery have been measured in triplicates and the stability have been measured in sextuplicate, which were expressed as mean and RSD. The content of 12 compounds have been measured in triplicates and expressed as mean.

The language expression can be improved for clarity of the information presented.

Response: Thank you for your advice. The language expression has improved carefully.